# A Systematic Review of the Heterogenous Gene Expression Patterns Associated with Multidrug Chemoresistance in Conventional Osteosarcoma

**DOI:** 10.3390/genes14040832

**Published:** 2023-03-30

**Authors:** Phakamani Goodman Mthethwa, Leonard Charles Marais, Veron Ramsuran, Collen Michelle Aldous

**Affiliations:** 1Department of Orthopaedic Surgery, Dr. Pixley Ka Isaka Seme Memorial Hospital, University of KwaZulu-Natal, 310 Bhejane Street, KwaMashu, Durban 4360, South Africa; 2Department of Orthopaedic Surgery, School of Clinical Medicine, College of Health Sciences, University of KwaZulu-Natal, 719 Umbilo Road, Durban 4001, South Africa; 3KwaZulu-Natal Research Innovation Platform (KRISP), University of KwaZulu-Natal, 719 Umbilo Road, Durban 4001, South Africa; 4Department of Clinical Medicine, School of Clinical Medicine, College of Health Sciences, University of KwaZulu-Natal, 719 Umbilo Road, Durban 4001, South Africa

**Keywords:** genetics, osteosarcoma, multidrug resistance, chemoresistance

## Abstract

Multidrug chemoresistance (MDR) remains the most significant obstacle to improving survival in osteosarcoma patients. Heterogeneous genetic alterations characterise the tumour microenvironment, and host molecular markers have been associated with MDR. This systematic review examines the genetic alterations of molecular biomarkers associated with multidrug chemotherapy resistance in genome-wide analysis of central high-grade conventional osteosarcoma (COS). We systematically searched MEDLINE, EMBASE, Web of Science, Wiley online library and Scopus. Only human studies involving genome-wide analysis were included, while candidate gene, in vitro and animal studies were excluded. The risk of bias of the studies was assessed using the Newcastle-Ottawa Quality Assessment Scale. The systematic search identified 1355 records. Following the screening, six studies were included in the qualitative analysis. There were 473 differentially expressed genes (DEGs) associated with chemotherapy response in COS. Fifty-seven of those were associated with MDR in osteosarcoma. The heterogeneous gene expressions were related to the mechanism of MDR in osteosarcoma. The mechanisms include drug-related sensitivity genes, bone remodelling and signal transduction. Complex, variable and heterogenous gene expression patterns underpin MDR in osteosarcoma. Further research is needed to identify the most relevant alterations for prognostication and to guide the development of possible therapeutic targets.

## 1. Introduction

Osteosarcomas are a group of aggressive primary malignant tumours of bone of mesenchymal cancer stem cell origin that produce osteoid matrices. They most commonly occur in children and adolescents [1]. Central high-grade conventional osteosarcoma (COS) is the most common subtype, comprising 90% of all osteosarcoma variants [1,2]. Currently, the treatment of COS typically involves neoadjuvant chemotherapy and wide surgical excision, followed by adjuvant chemotherapy [3,4,5]. The predictors of a poor prognosis regarding survival are male sex, older age, large tumour volume, non-extremity tumour, proximal long bone sites, poor response to chemotherapy, no surgical treatment, and amputations [3,4,5]. The tumour response to chemotherapy is the most important prognostic factor for predicting long-term survival. A good response is defined as >90% tumour necrosis (Huvos grade III/IV), and a poor response as < or =90% tumour necrosis (Huvos grade I/II), [3,4].

Despite advances in multidrug chemotherapy and surgical procedures, the 3-year event-free and 5-year survival of non-metastatic COS have plateaued at 60–70% [3,4,5]. Chemoresistance remains the most significant obstacle to improving long-term survival. The response to neoadjuvant chemotherapy has been poor in 40–45% of cases [5]. Neither tailored postoperative chemotherapy according to histological response to preoperative chemotherapy nor dose intensification has improved survival rates [5,6,7].

A heterogeneous gene expressions patterns characterise the tumour microenvironment, with a host of molecular markers associated with MDR [8,9,10]. Multidrug chemoresistance is defined as the innate and acquired ability of cancer cells to evade the effects of chemotherapeutics, mediated by mechanisms such as drug uptake and transport, detoxification in the cell, apoptosis inhibition, the repair of DNA damage, and osteosarcoma stem cells [11]. The mechanism of drug resistance is multifactorial and includes transporter pump disruption, oncogenes, tumour suppressor genes, abnormal DNA repair, mitochondrial alterations, autophagy, and epithelial–mesenchymal transition [11]. Osteosarcoma multidrug chemoresistance is typically investigated using microarray gene expression analysis [8,9,10]. Examining mutations across the genome, i.e., a genome-wide approach detects genetic instability, which could serve as biomarkers for the specific disease. Thus, it serves to find all mutations associated with MDR rather than focusing on the presence or absence of specifically selected abnormalities. Identifying MDR’s most relevant causes and biomarkers is needed for prognostication purposes and to guide treatment. Risk stratification using surrogate markers of chemotherapy response and disease burden in osteosarcoma is being predicted for the future [11]. Gene therapy targeting specific genes and molecules involved in chemoresistance is being widely researched globally [11].

This systematic review examines the heterogenous gene expression patterns associated with multidrug chemotherapy resistance in genome-wide analysis of central high-grade conventional osteosarcoma (COS).

## 2. Materials and Methods

### 2.1. Protocol and Registration

The study protocol, registered with Prospero (Registration No: CRD42021241510), specified the objectives, methods of analysis, and inclusion/exclusion criteria in advance. PRISMA (Preferred Reporting Item for Systematic Reviews and Meta-analyses) guidelines were used to design, conduct, and report the present systematic review [12].

### 2.2. Eligibility Criteria

We included all original research studies involving genome-wide analysis on human subjects with histologically confirmed high-grade central osteosarcoma. Studies involving other subtypes or variants of osteosarcomas or bone sarcomas were excluded. All prospective or retrospective experimental and epidemiological study designs were considered eligible, including randomised controlled trials, non-randomised controlled trials, quasi-experimental, cohort studies, case-control studies, and cross-sectional studies. Candidate gene studies and studies on animal models, and in-vitro studies investigating human tissue-derived cell lines were excluded. We also excluded all narrative reviews, case reports, congress proceedings, letters to the editor and opinion pieces. There was no limit set in terms of the publication date, and only articles published in English were included.

### 2.3. Search Strategy

In February 2023, a comprehensive literature search was performed on five online databases: MEDLINE, Scopus, EMBASE, Wiley online library and Web of Science. We also searched our archives, reviewed reference lists from identified articles and searched through the cited references of crucial publications. We used the following search terms to search the database: MeSH search terms used as per 2023 MeSH Descriptor Data: (“Genetic”) AND (“Osteosarcoma”) AND (“Multidrug resistance” OR “Chemoresistance”).

### 2.4. Study Selection

Two authors (PM and LM) independently screened all studies by title and abstract following de-duplication. All relevant articles were retrieved, and a second selection was performed on the same authors’ full-text paper version. After the final selection, publications were cross-referenced, and the two authors accounted for ambiguities.

### 2.5. Data Extraction

The custom spreadsheet for data analysis was created with Microsoft Excel. The data were extracted according to the author, country of origin (ethnicity), year of publication, journal, study design, number of participants, method of genetic analysis, chemotherapy protocol, primary study outcome measure, chemotherapy resistance pattern, rate of resistance to chemotherapy and gene or molecular marker associated with resistance.

### 2.6. Quality Assessment of Individual Studies

A risk of bias assessment was performed using the Newcastle-Ottawa Quality Assessment Scale by two independent reviewers (PM and LM). Each study was judged on three broad perspectives: selecting the study groups, comparability of groups, and ascertaining the outcome of interest. Each dimension was scored on a scale by allocating the total number of stars (9 for each study) [13] (Table 1).

### 2.7. Data Synthesis

A descriptive synthesis was carried out using each study’s extracted data and significant findings. The heterogeneity of the studies did not allow for quantitative data analysis. All gene symbols and name verification were thoroughly checked using GeneCards Human Database (https://www.genecards.org (accessed on 2 February 2023). 

## 3. Results

### 3.1. Study Selection

The electronic database search strategy identified 1355 publications. Three additional records were identified from other sources, bringing 1358 studies. Following de-duplication, 645 records were screened by title and abstract, of which 626 were excluded. Nineteen studies full-text articles were assessed for eligibility, and thirteen records were excluded: candidate gene, animal, experimental, blood cells, non-conventional osteosarcomas, non-English language, and review studies, (see Figure 1). Therefore, six studies were finally included for qualitative data synthesis.

### 3.2. Study Characteristics and Interventions

The study designs of the included studies and the methodology they employed are provided in Table 1. All six included studies were prospective non-randomised case-control studies. The country of origin was two Europeans, two Americans, one Japanese, and one Australian ethnicities. The sample size ranged from 8 to 52 patients, with 151 cases. Patient age ranged from 10 to 70 years and was not reported in two studies. In the studies that reported the age, 60 of 69 (87%) patients were below the age of 20 years. Chiappetta et al., and Endo-Munoz et al., included five patients with osteosarcomas of the axial skeleton, pelvis, or shoulder girdle. Almost all cases involved histologically confirmed high-grade conventional osteosarcoma. A single study included two patients with surface (periosteal) osteosarcomas. 

The chemotherapy protocols used were reported in all but one study [14]. The chemotherapy regimen was methotrexate, adriamycin, and cisplatin (MAP) with or without Ifosfamide (I) or etoposide in four studies. In comparison, the other two studies were chemo-protocol did not include methotrexate [10,15]. Only Endo-Munoz et al., reported the doses of the drugs, with doxorubicin, 25 mg m^−2^ and cisplatin, 100 mg m^−2^. In all studies, no further information was provided with the duration of chemotherapy. A poor response to chemotherapy was noted in 38% to 77% of cases. 

### 3.3. Study Designs and Outcome Measures

The response of the tumour to chemotherapy in each case was determined, in all studies, according to the Rosen protocol with a Huvos criterion of I or II (< or =90% tumour necrosis) defined as a poor response to chemotherapy and III or IV (>90% tumour necrosis) as a good response [3,4].Tissue samples for genetic profiling were obtained from the primary tumour’s initial biopsies before the chemotherapy initiation in all but one study. The gene expression profile of the primary tumours before chemotherapy in cases that subsequently exhibited a poor response to neoadjuvant chemotherapy was then compared to that found in subjects with a poor response. The study by Man et al. involved 28 patients and 34 tissue samples [9]. In this study, 14 samples were collected before the initiation of chemotherapy and 20 following chemotherapy at the time of definitive tumour resection. In six patients, samples were taken before and after chemotherapy. The authors hypothesised that samples taken after chemotherapy would enhance the detection of differences between chemo-resistant and chemo-sensitive tumours as the post-chemotherapy samples would contain more resistant cells than pre-chemotherapy samples. The authors identified 45 genes associated with a poor response to chemotherapy. They then used seven to build a predictive classification algorithm tested on pre-chemotherapy tissue samples. 

The gene expression profile associated with a poor response to chemotherapy was the primary outcome measure in five studies. In two studies, additional whole-genome sequencing (WGS) or whole-exome sequencing (WES) was also performed. The techniques used for genetic analysis are provided in Table 1 and include DNA or Oligonucleotide microarrays, exome probes, suppression subtractive hybridisation (SSH), and quantitative RT-PCR (Reverse Transcriptase Polymerase Chain Reaction). Microarray analysis is the most popular method of investigating chemoresistance in osteosarcoma. The microarray experiment starts with sample collection from which the targets are derived, followed by hybridisation of the target to the microarray probe and the elaboration and analysis of gene expression. However, each stage is subject to variability, namely tissue source, harvesting, platforms, sample labelling, data analysis and human error [17]. Meanwhile, the advantage of the WGS analysis of the entire coding genome could lead to a better understanding of the molecular mechanism underlying the development and progression of osteosarcoma. Furthermore, the WES approach could clarify the landscape of the genetic alteration in osteosarcoma and provide relevant biological data. In particular, this knowledge may result in an easier discovery of new proteins as molecular targets, which could be aimed at personalised therapies [18]. In contrast, SSH is a molecular biology technique that identifies DEGs between two groups with high sensitivity by comparing poor to good responders to chemotherapy among osteosarcoma patients [19].

### 3.4. Relationship of the Gene Expressions with Chemoresistance and Their Biological Mechanisms

A total of 473 differentially expressed genes were associated with response to chemotherapy in osteosarcoma. Fifty-seven were responsible for chemoresistance. The method by which the authors selected the genes associated with chemoresistance varied (Table 2). Endo-Munoz et al., Mintz et al., and Ochi et al. listed all DEGs and their average change in expression levels. Man et al. randomly selected seven from a pool of 45 significantly associated genes to measure mRNA expression, which was used to build their predictive model. Salas et al. selected two of 126 DEGs (STAT3, ERK1) based on their roles in tumorigenesis or chemoresistance for further analysis by QRT-PCR in an independent cohort. Chiappetta et al. selected 15 genes that were altered in non-responders and responders. The biological mechanisms of altered DEGs include extracellular matrix bone remodelling, signal transduction pathways, and drug sensitivity (Table 3).

Endo-Munoz et al. found 15 DEGs associated with chemoresistance, among the genes include TMSB10, metallothionein family members, GSTP1, and CYP4X1 [15]. Their analysis of osteosarcoma transcriptomes found several differentially expressed metallothionein family members and the deregulation of the genes involved in antigen presentation [15] The metallothionein family are involved in the anti-apoptotic chemoresistance pathway in osteosarcoma conventional drugs were also found by Mintz et al [8]. In contrast, the anti-apoptotic gene, BCL2, was found by Chiapetta et al. using the WES approach [14]. Endo-Munoz et al. tumours also exhibited a significant increase of ID I and profound correlation down-regulation of S1008, highlighting the potential as a therapeutic target for osteosarcoma [15] A similar TMSB10 gene was found by Endo-Munoz et al. [15]. In 2004, Ochi et al. identified it within the suite of DEG between good and poor responders in osteosarcoma tumour biopsies [10].

### 3.5. Genetic Alterations in Non-Responders and Biological Mechanisms

A total of 473 differentially expressed genes were associated with a poor response to chemotherapy in osteosarcoma. The biological mechanisms of altered DEGs include extracellular matrix bone remodelling, signal transduction pathways, and drug sensitivity (Table 2). In addition, both authors’ microarray analyses exhibited GSTP1 and CYP4X1 genes responsible for cell detoxification, a drug chemoresistance mechanism in osteosarcoma [10,15].

In this review, we also uncovered differentially expressed genes responsible for poor chemotherapy response of osteosarcoma, which was also involved in a drive toward osteoclastogenesis, extracellular bone matrix remodelling (ECM), and tumour progression. Using a microarray, Mintz et al. generated a genetic fingerprint of chemoresistance osteosarcoma tumours [15]. They were later validated via quantitative RT-PCR on desmoplakin, OPG, plasminogen activator inhibitor type 1, biglycan, annexin 2, PLA2G2A, and SPARC-1 [15]. These genes were dysregulated in osteoclastogenesis and ECM in poor chemo-response samples [15] The OPG, a secreted glycoprotein that regulates bone resorption through the inhibition of osteoclast differentiation, was also found using microarray analysis by other 2 authors (Ochi and Man et al.) Additionally, transforming growth factor beta 1 (TGF-ß1), also involved indirectly with osteoclastic activity, was expressed in two studies [9,11]. Annexin 2 (ANXA2), another gene indirectly involved with the osteoclastic activity and bone resorption, was significantly expressed in poor chemotherapy response of osteosarcoma patients [9,15].

The osteosarcoma chemoresistance genes are responsible for signal transduction pathway and angiogenesis, ERK1 and STAT3 mRNA expression significantly correlated with poor response to chemotherapy when analysed by suppression subtractive hybridisation (SSH), validated by QRT-PCR and immunohistochemistry analysis [15]. Furthermore, Salas et al. authors suggested elevated predictive value of a high score of both pSTAT3 and pERK1 in combination (90%) could be used as a surrogate marker for diagnostic purposes [15] Another signal transduction pathway and angiogenesis gene group found in poor chemotherapy response in osteosarcoma patients included ERBB4/HER2, AKR1C4, TWIST1, TMPO, MCM2, FGFR1, and PTN [8,9,10,14,16].

## 4. Discussion

This systematic review has explored the heterogenous gene expression patterns associated with multidrug chemoresistance (MDR) in high-grade conventional osteosarcoma. We summarized six genetic studies in humans that interrogated the relationship between osteosarcoma and response to chemotherapy. All included studies in this review have defined cohort clinical characteristics, response to chemotherapy according to Huvos grading system, and gene expression profiles. Firstly, our systematic approach uncovered that conventional osteosarcoma exhibits a genetic signature consistent with a chemo-resistant phenotype. Secondly, we revealed 473 genes or molecular markers associated with the response to osteosarcoma chemotherapy. Of which 57 chemoresistance genes were found, 15 were DEGs in non-responders, 15 were DEGs with the most significant fold change high expression levels, and the rest were randomly selected or based on their roles in chemoresistance by authors. The genes mentioned above are altered in the biological mechanisms of conventional chemotherapy drug sensitivity, extracellular bone matrix, and signal transduction pathways.

### 4.1. Drug Sensitivity Chemoresistance Genes

Surprisingly, the genes or molecular markers identified by the six included studies were inconsistent, with no genes identified by more than two studies. Furthermore, there was significant variation and no concordance in the over or under-expressed genes correlating with chemoresistance in high-grade osteosarcoma. Genes or molecular markers that are classically associated with MDR to conventional therapies of osteosarcoma were not found [10]. For instance, drug sensitivity genes, including ABC transporters, block the binding of chemotherapy drugs, and DNA repair genes were not differentially expressed [10]. Specifically, P–pg, a transmembrane ATP-dependent efflux pump encoded by the MDR1 gene, may lead to chemoresistance, poor outcomes, and an increased frequency of adverse events in osteosarcoma [10,20,21,22]. Chemoresistance mediated by the MDR1 gene involves, among other agents, doxorubicin, the treatment drug for osteosarcomas [20,21,22]. Furthermore, P–pg. overexpression is associated with cisplatin efficacy in osteosarcoma patients [23].

In contrast, the anti-apoptotic gene BCL2 and metallothionein family (MT 1G and 1L) were upregulated in high-grade osteosarcoma according to work using different genetic analysis methodologies (Table 3) [14,15]. Chiappetta et al. reported BCL-2 among the DEGs associated with poor responses in osteosarcoma [14]. However, the studies included in this review did not find genes in the same family of tumour suppressors and anti-apoptotic genes. For instance, TP53 and BAX are considered in osteosarcoma, and their overexpression promotes tumorigenesis, progression, and resistance [24].

The metallothionein family of genes classically responsible for intrinsic and acquired MDR were revealed using microarray analysis in osteosarcoma [8,15]. Moreover, two genes, GSTP1 and CYP4X1, involved in cell detoxification (MDR mechanism) were found to be overexpressed using microarray analysis by two authors [11,15]. GSTP1 gene overexpression has been associated with poor response in osteosarcoma and other cancers [11].

### 4.2. Extracellular Bone Matrix Chemoresistance Genes

The studies included in this systematic review further revealed that the osteosarcoma microenvironment displays gene signatures that impaired osteoclast genesis and enhancement of chemoresistance; however, variations of DEGs also existed [8,15]. The decrease in expression of genes OPG, PLA2G2, TGF-ß1 and annexin 2 confers bone resorption, osteoclastogenesis, and poor response to chemotherapy [8,15]. Thus, the osteosarcoma microenvironment is saturated with gene signatures responsible for the chemoresistance phenotype. 

### 4.3. Signal Transduction Pathway Chemoresistance Genes

Again, inconsistencies existed in gene signatures responsible for osteosarcoma cell proliferation, angiogenesis, invasion, and metastasis. (Table 2). Using different genetic analyses in osteosarcoma tissues, the authors uncovered the genes implicated in chemoresistance signal transduction pathways [8,11,14]. ErbB4 and STAT3 confer osteosarcoma chemoresistance via the inhibition of apoptosis [14,16,25,26]. In contrast, PTN expression was increased 3.2-fold in poor chemotherapy responders in osteosarcoma [8]. Furthermore, high PTN expression has previously been shown to confer a poor prognosis in osteosarcoma [27]. However, future research is warranted on the genes mentioned above.

### 4.4. Strengths and Limitations

To our knowledge, this is the first systematic review summarising the gene expression profiles associated with multidrug chemotherapy resistance in human conventional osteosarcoma. Recent reviews focused mainly on mechanisms of MDR, microRNA, candidate gene association, and genetic polymorphism reviews. Moreover, the strengths of this review include systematic search using stringent criteria, using explicit keywords previously validated, five databases, applying study design according to PRISMA guidelines, and using data synthesis of all eligible genetic association studies.

However, some discordances in the studies are the ethnicity disparities, which have mainly focused on the Caucasian and Asian populations, with a lack of evidence for individuals of African descent. [8,9,10,11,28] Moreover, selection bias is associated with the studies’ sampling strategy. Owing to the rarity of osteosarcoma, the small sample size and lack of validation remain major shortcomings in investigating this cancer. Furthermore, inconsistencies were noted in the study designs, with the clinical characteristics including age, sampling strategy, the inconsistent reporting of chemotherapy protocols or drugs, doses, response to chemotherapy, improper standardised experimental protocols, the use of different technologies for data acquisition and analysis are potential shortcomings in this review. The relationship between gene expression and chemotherapy response in some included studies must be appropriately defined. Therefore, different chemotherapy strategies indirectly affect our results. Thankfully, the EURAMOS-1 clinical trial has addressed these issues [5]. Additionally, the microarray technology used in various investigations can reveal significant variability in gene expression subject to experimental conditions. The variability in the laboratory techniques used in each study makes comparability difficult and may even show conflicting results. Furthermore, the studies involved analyses of pre-selected targets producing another selection bias. Because of this, our systematic review represents pre-selected expression analyses rather than an accurate representation of the genome-wide expression profile.

### 4.5. Future Directions

The uncovered COS chemoresistance genes or molecular markers and mechanisms pathways deserve further investigation. Furthermore, we highlighted potential surrogate markers and therapeutic targets underpinning chemotherapy response in COS. Wider gene expression analyses involving international collaboration may be the key to unlocking many new potential prognostic and treatment guidance. Thus, owing to the rarity of COS, we recommend the routine storage of the biopsy specimens for future genetic analyses. The slight increase in COS among Africans suggests unknown risk factors and genetic alterations [29]. Therefore, future research should focus on a global approach.

## 5. Conclusions

This systematic review of gene expression has consolidated the knowledge of several candidate genes/molecular markers associated with chemoresistance in osteosarcoma. Additionally, the genes mentioned previously are involved in drug sensitivity, bone remodelling, and signal transduction. The identified molecular markers warrant further research to unlock the chemoresistance of osteosarcoma. Osteosarcoma is a rare cancer; further international collaborative work is needed, and large cohort studies are necessary to validate these findings, and these could identify biomarkers for clinical prognosis and drug targeting.

## Figures and Tables

**Figure 1 genes-14-00832-f001:**
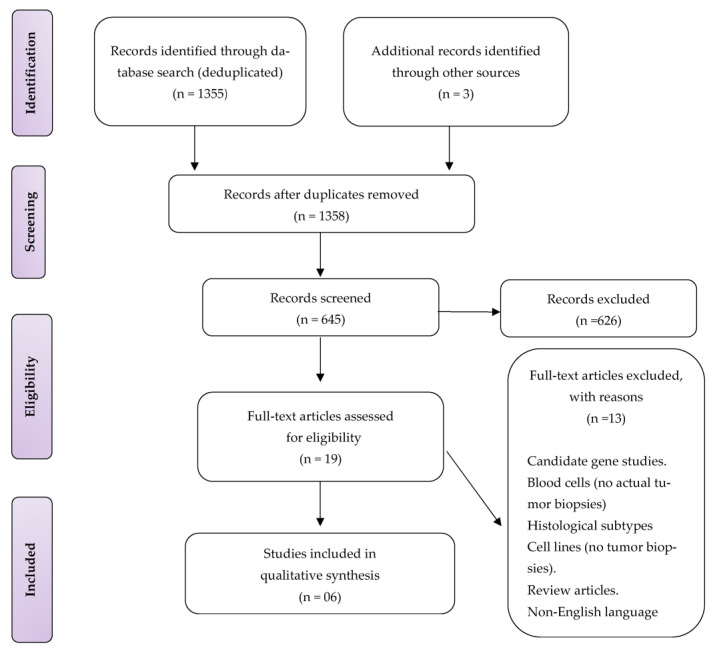
PRISMA flow diagram showing a selection of included studies.

**Table 1 genes-14-00832-t001:** Study characteristics and interventions.

Author	Country	Sample Size	Age (Range)	Patient Age	Sampling Strategy	Non-Conventional Osteosarcomas	Chemotherapy Drugs and Doses	Response to Chemotherapy	Genetic Analysis	Method	Newcastle-Ottawa Score
Chiappetta et al. [14]	Italy	8	11–35	2/8	Primary tumour before chemotherapy	0	N/A	38%	* WES + GEP	NGS (Exome probes) + Quantitative RT-PCR	7
Endo-Munoz et al. [15]	Australia	22	7–67	16/22	Primary tumour before chemotherapy	0	* Doxorubicin 25 mg m^−2^ andCisplatin 100 mg m^−2^	77%	* WGS + GEP	Oligonucleotide Microarray + RT-PCR	9
Man et al. [9]	USA	20	11–21	19/20	Primary tumour before and after chemotherapy	0	* High-dose methotrexate, doxorubicin, and cisplatin	64%	* GEP	cDNA Microarray + Quantitative RT-PCR	7
Mintz et al. [8]	USA	30	-	-	Primary tumour before chemotherapy	0	* High-dose methotrexate, doxorubicin, and cisplatin	50%	* GEP	Oligonucleotide Microarray + Quantitative RT-PCR	9
Ochi et al. [10]	Japan	19	10–70	13/19	Primary tumour before chemotherapy	2	* Doxorubicin 90 mg m^−2^,Cisplatin 120 mg m^−2^, ifosfamide 15 g m^−2^	54%	* GEP	cDNA Microarray + Semi-Quantitative RT-PCR	7
Salas et al. [16]	France	52	17.4 (mean)	-	Primary tumour before chemotherapy	0	* High-dose methotrexate, doxorubicin	54%	* GEP	cDNA SSH + Quantitative RT-PCR	7

* Methotrexate, Adriamycin and Cisplatin (MAP) or Ifosfamide (I). * Whole-Genome Sequencing (WGS) * Whole-Exome Sequencing (WES). * New Generation Sequencing (NGS). * Gene Expression Profile (GEP).

**Table 2 genes-14-00832-t002:** **Fifty-seven** genes associated with a poor response to chemotherapy.

Study	Number of DEGs	DEGs Associated with Non-Response
Chiappetta et al., 2017 [14]	15i	ALDHIL2, BCLAF1, CLCN1, COG3, DIS3, ERB4, KARS, OR52N1, PDE6C, PDHX, SCN8A, SP140L, THBS1, UBE4A, ZNF12
Endo-Munoz et al., 2010 [15]	123ii	TMSB10, SPP1, IFI30, RPS2, HLA-B, STK11, CTSB, IL17RC, FTH1, CFL1, HLA-A, TYROBP, RPS19, RPS20, RPS 29
Man et al., 2005 [9]	45iii	TWIST1, PDCD5, OXCT, TMPO, UBE2A, EFNB2, AMPD2
Mintz et al., 2005 [8]	104ii	KRT19, PLA2G2A, SPARCL1, DSP, TPS1, GAPD, RAB4B, CLDN5, THBS4, TNFRSF11, F13A1, STC2, SCYA14, FGF2, VWF
Ochi et al., 2004 [10]	60iii	AKR1C4, GPX1, GSTTLp28,
Salas et al., 2014 [16]	126iii	STAT3, ERK1

(i) Fifteen DEGs only occurring in non-responders listed, (ii) Fifteen DEGs with greatest change in expression level listed, and (iii) Randomly selected by authors.

**Table 3 genes-14-00832-t003:** Differentially expressed genes (DEG’s) according to biological mechanism in high grade osteosarcoma.

Biological Function	Genes Symbol	Gene Name	References
Extracellular Matrix Bone Remodelling (Osteoid Formation and Resorption)			
	OPG TGF-ß1 PLA2G2A TREM2 DSP ANXA2	Osteoprotegerin Transforming Growth Factor beta 1 Phospholipase A2 Group IIA Triggering receptor expressed on myeloid cell. Desmoplakin Annexin A2	[8,15] [8,10,15] [8] [8] [8] [8,15]
Signal transduction pathways and angiogenesis			
	ERBB4/HER2 STAT3 AKR1C4 TWIST1 TMPOMCM2 FGFR1 PTN	Erb-B2 Receptor Tyrosine Kinase 4 Signal Transducer and Activator of Transcription 3 Aldo-keto Reductase Family 1 Member C4 Twist Family BHLH Transcription Factor 1 ThymopoietinMinichromosome Maintenance Complex Component 2 Fibroblast Growth Factor Receptor 1 Pleiotrophin	[14] [16] [10] [9] [9] [9] [9] [8]
Drug sensitivity			
Anti-apoptotic	BCL-2 MT1G MT1L	BCL2 Apoptosis Regulator Metallothionein 1G Metallothionein 1L, pseudogene.	[14] [8,15] [8,15]
Detoxification in cell	CYP4X1 GSTP1	Cytochrome P450 Family 4 Subfamily X Member 1 Glutathione S-transferase Pi 1	[10,15] [10]

## Data Availability

No new data was generated.

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
