# Peer review of "A Systematic Review of the Heterogenous Gene Expression Patterns Associated with Multidrug Chemoresistance in Conventional Osteosarcoma"

_genes, 2023, doi:10.3390/genes14040832_

Round 1

Reviewer 1 Report

This review can be considerately improved. I have doubts the way it was designed is of interest to the journal. It looks more like a brute force data review than a metaanalysis or a review with the purpose to reach a useful conclusion. I would improve it by aiming more complex analysis and digging deeper into the data. Just a compilation of papers does not justify a high quality publication.

Reviewer 2 Report

Authors seemed to conduct systematic review on gene expression associated with chemoresisteance in osteosarcoma. This is an interesting topic, but the manuscript has some points to be improved for publication. Dataset has not been described in details from each study. I also believe that more relevant and recent literature should be cited to make this work up-to-date and reliable. Additional detailed comments are as follows.

Line 25-27. Following the screening, six studies were included in the qualitative analysis. There were 473 differentially expressed genes (DEGs) associated with chemotherapy response in COS. Fifty-seven of those were associated with MDR in osteosarcoma.

It is essential for a excellent systematic review to add recent data from the latest research after you initially completed finished your work on January 2022.

Line 41-44. Tumour response to chemotherapy has emerged as probably the most important prognostic factor for predicting long-term survival. A good response is defined as > 90% tumour necrosis and a poor response as <or = 90% tumour necrosis. [3,4]

It will support readers to describe prognostic factors other than the response to chemotherapy in osteosarcoma.

Line 89. In January 2022, a comprehensive literature search was performed on five online databases:

It should be requried to search for more recent studies after January 2022.

Line 94. AND "Multidrug resistance

I suggest to use “OR “Chemoresistance” or other search terms about drug resistance.

Line 100-105. Data extraction

You should present more details such as specific names of regimens, not just chemotherapeutic protocol, with resistance and their relationship with genetic expressions.

Table 1

More details such as duration, doses, and intervals between each regimen are required.

Line 190-193. “The techniques used for genetic analysis are provided in table 1 and include DNA or Oligonucleotide microarrays, exome Probes, Suppression Subtractive Hybridisation (SSH), and Quantitative RT-PCR (Reverse Transcriptase Polymerase Chain Reaction)."

Pros and Cons of each technique should be added.

Discussion

It should be mentioned whether the difference of cemotherapy protocol in each study affected your study results or not.

Line 220-222. Surprisingly, the genes or molecular markers identified by the six included studies were inconsistent, with no genes identified by more than two studies.

As you mentioned inconsistency and no common genes identified, it is well known that genetic mutations of the osteosarcoma are heterogenous and complex. Therefore, it seems to have little benefit to collect and analyze the pattern of gene expression. 

Reviewer 3 Report

The review paper by Dr Mthethwa and colleagues presents a potentially very interesting and useful analysis of gene expression profiles related to multidrug resistance in OS. In addition to the heterogenicity of genetic alterations, appropriately described in the manuscript, what stands out from reading this paper are the challenges (both technical and methodological)  that affect such an approach in OS. Just to list a few: i) OS is rare cancer; ii) poorly standardized experimental protocols; iii) the use of different technologies for data acquisition and analysis. All the issues/challenges should be clearly addressed and discussed.  

Author Response

  1. Point 1. Please see the attachment.
  2. The paper will be sent for English Editing.  

Round 2

Reviewer 3 Report

The manuscript has been revised according to this Reviewer's comments and it is now acceptable for publication